# Factors Associated with Compliance among Methadone Maintenance Treatment Transfers: Evidence from Audit Records at Clinics in Guangdong, China

**DOI:** 10.3390/ijerph16112023

**Published:** 2019-06-06

**Authors:** Cheng Gong, Xia Zou, Wen Chen, Yin Liu, Qian Lu, Li Ling

**Affiliations:** Department of Medical Statistics, School of Public Health, Sun Yat-sen University, Guangzhou 510080, China; gongch25@mail2.sysu.edu.cn (C.G.); zouxia@mail3.sysu.edu.cn (X.Z.); chenw43@mail.sysu.edu.cn (W.C.); liuy429@mail2.sysu.edu.cn (Y.L.); luqian7@mail2.sysu.edu.cn (Q.L.)

**Keywords:** methadone maintenance treatment, opioid dependence, transfer, service, delayed treatment

## Abstract

Methadone maintenance treatment (MMT) requires patients to intake their daily dose in person at their clinic. Therefore, transfer services are vital for patients who need temporary leave from their primary MMT clinic. However, studies have shown that transfer patients might delay return after temporary leave, leading to missed doses and putting them at risk of increased harm. In this study, we aimed to explore the transfer rates and factors associated with MMT patients who delayed return during a transfer period. In this retrospective analysis, we used audit records from the web-based management system from six MMT clinics in Guangdong, China. Multilevel logistic regression and multilevel Poisson regression analyses were used to examine the factors associated with patients who delayed return to their primary MMT clinic. A total of 459 people used the transfer system 2940 times between January 2006 and December 2016. Of those, patients delayed return to their primary MMT clinic 1199 times (40.78%). Patients who transferred regularly had poor compliance rates with MMT treatment. Those who once dropped out from and then re-enrolled in MMT were more likely to delay return. Most patients (82.71%) who used the transfer service for “work” were more likely to prolong their delay length. The findings highlight that a more flexible transfer system would minimize inconvenience to the patients.

## 1. Introduction

Opioid dependence is a chronic health condition that necessitates long-term treatment [1,2]. Methadone maintenance treatment (MMT) is regarded by the Word Health Organization (WHO, 2009) as the most effective substitute therapy for opioid addiction. According to the pharmacological action of methadone, daily treatment is needed to minimize a user’s withdrawal symptoms [3].

With long-term treatment, the social functioning of MMT patients improves greatly [4]. With the increase in social activities such as work and travel, patients’ demands for temporary leave gradually increase [5,6]. In response, take-home doses and transfer services are the most prevalent solutions worldwide. For example, in Singapore, MMT clinics provide take-home doses for the patients who are unable to attend the clinic every day [7]. However, the take-home doses are not available for all patients in the clinic [8]. Take-home doses are a privilege and are only provided to those patients who (1) have fully cooperated with the treatment, and (2) have a minimum of three months with no positive urine drug test [8]. Additionally, empirical evidence has shown that take-home doses, without strict medical supervision, could increase the risk of misuse and diversion, such as methadone injection and methadone being sold out under the black market [9,10,11,12].

Different from take-home doses, some countries, such as Australia, Myanmar, and China, provide transfer services to patients who need temporarily leave from their primary clinic. As a fundamental aspect of care, transfer services allow transfer patients to continue their treatment at another MMT facility near their destination [13,14,15,16]. The transfer patients are able to take their doses under medical supervision without the potential risk of misuse and diversion. When the patients need to use the transfer service, they are required to report their duration of leave, destination, and reason for transfer request in their transfer form for each transfer request [13,17,18] so that the administration can arrange the transfer with their treatment history, including the details of treatment date and dose [18,19]. This transfer form is delivered from the primary clinic to the receiving clinic through manual submission or an internet management system. China has the world’s largest methadone service system [20]. The transfer service system has been running more than 10 years in China. When the patients need to take a temporary leave of absence from their primary registered MMT clinics, they can theoretically continue their treatment at another MMT facility near their destination [16]. Transfers are arranged in advance to avoid interrupted treatment and minimize inconvenience to the patients.

A previous study showed that patients who use the transfer service commonly delay return to their primary MMT clinic after temporary leave [21]. Delayed return is identified as when the patient returns later than the date declared in a patient’s transfer application to their primary MMT clinic. About half of MMT patients who transfer delay their return to their primary MMT clinic by three days on average [21,22]. When patients delay their return to their primary MMT clinic, they miss doses and might be at increased risk of withdrawal symptoms, drug relapsing, and spreading infectious diseases [23,24,25,26]. However, to the best of our knowledge, no study has ever assessed association between transfer service and patients’ delayed return.

To fill this gap, we first examined the participant transfer rates. Second, we explored the factors associated with patients who are likely to delay their return to their primary MMT clinic.

## 2. Materials and Methods

### 2.1. Study Site and Sampling

In Guangdong province, South China, there were 457,000 registered drug addicts in 2017, accounting for 17.9% of China’s total 2,553,000 reported drug addicts [27]. Due to the large number of registered drug users in Guangdong province, 66 MMT clinics were open at the end of 2016 [28]. Since 2008, all the clinics have provided transfer services. In this retrospective analysis, we invited all 66 MMT clinics in Guangdong to participate in this study. Six clinics in four cities across Guangdong province: Guangzhou, Dongguan, Jiangmen, and Yangjiang with full audit records all located in the Pearl River Delta agreed to join our study. The six clinics all opened in 2006, treating approximately 100 patients per day, and provided the same methadone maintenance treatment.

Patients who successfully used the MMT transfer service in the six MMT clinics from 1 January 2006 to 31 December 2016 were included in our study. Successful use of the MMT transfer service was defined as the patient having records about the transfer application and medical records after returning to their primary MMT clinic. We excluded (1) patients whose primary clinic was not one of the six sites included in this study and (2) patient dropouts, defined as those who missed 14 continuous days of treatment.

### 2.2. Data Collection

The Chinese National MMT Data Management System was established simultaneously with the opening of the first eight MMT clinics in China in 2004 across the provinces of Sichuan, Yunnan, Guizhou, Zhejiang, and Guangxi [11]. For medical supervision needs, each patient registered in a MMT clinic must have a unique number to access treatment. Each MMT clinic updates their service records according to the patient’s daily treatment. All daily treatment records were collected from the web-based Chinese National MMT Data Management System from the six clinics. We extracted eligible records between 1 January 2006 and 31 December 2016. The information in the records included: (1) characteristics of MMT patients (age, sex, education, and employment status); (b) methadone treatment-related information (start date of methadone maintenance treatment, number of times enrolled in MMT, the date of methadone intake, and dosage); (c) drug use history (age of initiated drug use, intravenous drug use, and the results of urine drug tests); and (d) transfer service records (the frequency of transfers, the start date of each transfer, the declared return date, and the actual return date of the patient, methadone intake and dosage during transfer, and the reason for transfer). All the records were collected and de-identified before analysis.

### 2.3. Study Variables

Dependent variables included delayed return (yes/no) and the length of delay. Delayed return was defined as the patient returning later than the date declared in their transfer to their primary registered MMT clinic. The length of delay (days) was the duration between the declared date of return in the patient’s application form and the actual date of return.

Our main independent variables were the characteristics of each transfer: (1) the frequency of transfer service use (times/year); (2) the duration of each transfer to other MMT clinics, which was defined as the duration from the date the transfer started to the declared return date in the patient’s application form; (3) the duration of MMT before transfer, which was calculated by the transfer start date minus the first MMT-enrollment date (year); (4) the average daily methadone dosage (mg/day) during the transfer period; (5) the reason for transfer (work, travel, medical, or other); (6) the compliance rate during the transfer period, which was calculated as the number of days of methadone intake over the number of applied transfer days; and (7) the result of urine drug tests in the past three months before transfer (%). We also included demographic characteristics (age, sex, employment status, and education level), methadone treatment-related characteristics (MMT dropout and re-enrollment rates, the duration of MMT before transfer), and drug use history (the results of urine drug tests in the past three months before transfer) as probable confounders in this study. 

### 2.4. Statistical Analysis

We used mean and standard deviation (SD) to describe normally distributed continuous variables. The median and interquartile range (IQR) were used to describe non-normally distributed continuous variables. The hierarchical structure of the data resulted in the clustering of transfers (level 1) among different patients (level 2). A two-level hierarchical logistic regression analysis was used to explore the risk factors of delayed return (Model A). The odds ratio (OR) and adjusted odds ratio with a 95% confidence interval (CI) of logistic regression are reported. We conducted a two-level hierarchical Poisson regression analysis to identify factors associated with the length of delay (Model B). Prevalence ratio (PR) and adjusted prevalence ratio with a 95% CI of Poisson regression are reported. A cluster effect was found at the individual level when an empty model was used (Model A: Intraclass Correlation Efficient (ICC) = 0.3581, *p* < 0.0001, Model B: ICC = 0.3452, *p* < 0.0001). Variables with bivariate *p*-value <0.10 in univariable regression were forced into subsequent multivariable regression. All the data analyses were performed using SAS, version 9.4 (SAS Institute Inc., Cary, NC, USA).

### 2.5. Ethical Statemtent

The study protocol was reviewed and approved by the Institutional Review Board (IRB) of the School of Public Health, Sun Yat-sen University, Guangzhou, China (No: 2013-26).

## 3. Results

### 3.1. Demographics and Methadone Treatment-Related History

From the data collected in this study, a total of 459 individuals completed 2940 transfers between 1 January 2008 and 1 December 2016. The mean age of the patients was 40.40 (SD = 6.22) years old, and most of them were male (88.45%), educated to middle school level (80.83%), and unemployed or had a part-time job (65.58%). The mean duration of methadone treatment was 6.80 years (SD = 2.03). The average initiated age of drug use was 23.12 (SD = 5.59) years old. Most patients had an intravenous drug use history (88.89%), and more than half of the patients (54.03%) had dropped out and re-enrolled in MMT (Table 1).

### 3.2. Transfer Patient Characteristics

The frequency of patient transfers each year varied from 1 to 32 times (Median: 0.78, IQR: 0.29–2.00). The average number of days documented on a patient’s application form for a transfer request to their secondary clinic was 9.00 days (IQR: 2.00–30.00). Most patients declared work as the reason for their transfer request (82.71%). Among all records, 40.79% of patients delayed returning to their primary MMT clinic after being transferred by an average delay of 3.00 days (IQR:1.00–6.00). Of the patients, 22.96% had at least one failed urine drug test in the last three months prior to transfer, and 18.03% of patients refused to do a urine drug test before being transferred (Table 2).

### 3.3. Factors Associated with Missed Doses in Delayed Return Period

Of the 2940 transfers, 1199 (40.79%) were categorized as delayed return. As shown in Table 3 (Model A), patients who had a poor compliance rate during the transfer period, e.g., missed doses while at a secondary clinic (adjusted OR: 0.16, 95% CI: 0.13–0.21, *p* < 0.0001), had a shorter duration of methadone treatment before being transferred (adjusted OR: 0.88, 95% CI: 0.81–0.95, *p* = 0.0017), and had previously dropped out of the MMT program (adjusted OR: 1.70, 95% CI: 1.21–2.41, *p* = 0.0022) were more likely to delay their return after being transferred (Table 3, Model A).

### 3.4. Factors Associated with the Length of Missed Doses in Delayed Return Period

The average length of delay was 3.00 days (IQR: 1.00–6.00) among 1199 delayed transfers. Patients who frequently transferred each year (adjusted PR: 1.06, 95% CI: 1.02–1.11, *p* = 0.0104) and those with a dropout and re-enrollment history (adjusted PR: 1.78, 95% CI: 1.38–2.29, *p* < 0.0001) were more likely to extend their delayed return to their primary MMT clinic. Patients who had poor compliance rates during the transfer period (adjusted PR: 0.26, 95% CI: 0.24–0.29, *p* = 0.0037), shorter duration of methadone treatment before transfer (adjusted PR: 0.89, 95% CI: 0.84–0.95, *p* = 0.0032), used the transfer service for work (adjusted PR: 1.20, 95% CI: 1.07–1.35, *p* = 0.0099), and had at least one positive urine drug test in the last three months before being transferred (adjusted PR: 1.10, 95% CI: 1.02–1.19, *p* = 0.0396) were more likely to extend their length of delayed return (Table 3, Model B).

## 4. Discussion

In the six MMT clinics included in our study, patients who used the transfer system regularly delayed return to their primary MMT clinic. Almost 40% of patients delayed return to their primary clinic by an average of three days. These findings are consistent with others [21] reporting that patients delayed return to their primary MMT clinic in Shenzhen by 3.55 days. According to Chinese regulations, patients cannot access MMT without the permission of their primary clinic if their transfer application expires while at a secondary clinic [13]. This regulation is said to protect patients from the risk of misuse and overdoses [11,29]. While reasonable, our research suggests that patients with a poor compliance history might be involved in a cyclical problem exacerbated by the administration system.

When a patient wants to extend their stay at their transfer destination beyond what was detailed in their transfer application form, they need to inform their primary MMT clinic. Then, the application needs to be processed and authorized by the patient’s primary MMT clinic’s doctor. It must also be approved by the clinic’s administrative regulators. In our experience from working in a clinic, it can take between one to two days for the National MMT Data Management System to be updated. This delay might push the patient to choose illicit drug use during the missed dose period(s), which puts them and others at increased risk of social harm [30]. We suggest that the MMT administrative system needs to be more flexible, so patients can more easily extend treatment at their secondary clinic. Given the immediacy of an online system, this would appear to present few information technology (IT) challenges as long as system checks are put in place.

Research has shown that MMT patients are self-employed or have a part-time job [31,32]. Most of the patients in this study declared “work” as their reason for transfer request (82%). This is much higher than reported in other research projects (40–50%) [33,34]. Notably, the participants who used the transfer service for work were more likely to extend their length of delay. This suggests that patients are struggling to balance their work and clinic commitments and may need to travel for work due to limited opportunities and potentially discrimination [35]. However, the data in our study concerning employment status were self-reported and collected at the time of enrolment, so it may not be reliable as most of the patients claimed they were not employed in the audit records (Table 1). Further in-depth research is needed as to why patients transfer and why they delay return to their primary MMT clinic.

Consistent with the research on urine drug tests from MMT patients, nearly 20% of patients failed a urine drug test within three months before being transferred [36,37], which is fewer than the 46% detailed in other research [38]. Even considering the number of patients who refused a drug test (18.03%) prior to being transferred, we are reasonable to think that transfer patients self-manage well during the transfer period.

A shorter patient MMT history and dropout and re-enrollment rate in MMT are general indicators of poor compliance with the transfer treatment. This finding is in agreement with a number of studies that have shown that patients in their early stages of treatment are more likely to contravene the clinic regulations [39,40,41]. Overall, delayed return is associated with the characteristics of patients and the MMT clinic transfer system. Patients with a dropout and re-enrollment history and shorter methadone treatment history were more likely miss their doses after temporary leave. We suggest changes need to be made to the administrative system to accommodate patients who transfer regularly. Our research also provides a reference for establishing a cross-regional system for MMT clinic transfer patients.

The findings of this study suggest that a few policy changes are needed. Firstly, an effective and flexible transfer management, especially the end-date of transfer, should be introduced to MMT systems to prevent delayed return. The authorities should provide convenient and quick transfer services for patients to avoid missed doses. Secondly, the coverage of MMT clinics and the comprehensive MMT services should be provided for all patients to improve participant compliance. Especially for patients with a drop-out history, we suggest closely monitoring their methadone dosage and urine drug tests to prevent the delayed return after transfer. Thirdly, for compliant participants with long-distance transfer, take-home dosage could be provided for two or three days to ensure they receive their dosage during the delay period. The take-home MMT strategy available in countries such as Israel, Singapore, and the United States has been effective in improving patient compliance [7,42,43,44]. In 2017, an innovative study on methadone take-home strategies was conducted in Yunnan, China, where the local authorities provided take-home doses for MMT patients using a high-tech internet-connected treatment box with four days’ dosages [45,46]. Through the internet-connected box, patients took methadone remotely via the supervision of the staff while physically absent from their primary MMT clinic [46]. Though this take-home, high-tech strategy may be widespread in the future, we suspect such a roll out is unlikely anytime soon. In the meantime, we should focus on improving compliance rates among MMT patients. Of particular concern to us here is the compliance rates among those who use the transfer system.

This present study has some limitations. One, information was exacted from the electronic database, meaning other factors may potentially be correlated with the delayed return that were not collected. Therefore, a future quantitative study is needed to investigate additional details about the transfer period. Two, all subjects were selected using convenience sampling from six clinics in Guangdong, which might lead to selection bias, so we need to be cautious about generalizing the findings to all MMT patients. Future investigation among transferred patients on a larger scale is needed to determine the long-term impact of delayed return on transfer experiences. Three, compared with other provinces of China, Guangdong province has better economic development and provides more jobs, which may affect the utilization of transfer service in the MMT. In further study, the investigation of features of MMT patients in Guangdong and other areas of China is needed.

## 5. Conclusions

In our analysis of six MMT clinics in Guangdong, China, delayed return among MMT patients who used the transfer system was common. Patients with a poor compliance history appear to be involved in a cyclical problem exacerbated by the fixed administrative system. Our study highlights the characteristics of transfer patients and identifies those at high risk of delayed return: those who have enrolled multiple times, those who have a short history of MMT, and those with a history of failed urine drug tests in the three months prior to transfer. Our research outlines the need for flexibility within the MMT administrative system in China to help increase compliance rates.

## Figures and Tables

**Table 1 ijerph-16-02023-t001:** Individual characteristics and methadone maintenance treatment (MMT) history (level 2).

Characteristics	*n* = 459
Demographics	
Sex, *n* (%)	
male	406 (88.45)
female	53 (11.55)
Age, Mean ± SD	40.41 ± 6.22
Employment status, *n* (%)	
unemployment or part-time job	301 (65.58)
full-time job	158 (34.42)
Education level, *n* (%)	
≤high school	371 (80.83)
>high school	88 (19.17)
Drug-related history	
Intravenous drug use, *n* (%)	
yes	408 (88.89)
no	51 (11.11)
Methadone treatment-related history	
Duration of received methadone treatment (years), Mean ± SD	6.80 ± 2.03
Dropout and then MMT re-enrollment, *n* (%)	
yes	248 (54.03)
no	211 (45.97)
Age of initiated drug use, Mean ± SD	23.12 ± 5.59
No. of times of transferred, interquartile range (IQR)	4.00 (2.00, 11.00)
Frequency of transfer service utilization (times/year), IQR	0.78 (0.29, 2.00)

**Table 2 ijerph-16-02023-t002:** The characteristics of each transfer (level 1).

Characteristic	*n =* 2940
Delayed return, *n* (%)	2940 (100)
yes	1199 (40.79)
no	1741 (59.21)
Average length of delay, IQR	3.00 (1.00, 6.00)
No. of days transferred to other MMT clinics, IQR	9.00 (2.00, 30.00)
Average daily methadone dosage during transfer period (mg/day), IQR	63.90 ± 33.44
Compliance rate during transfer period, IQR	0.75 (0.30, 1.00)
Reason for transfer, *n* (%)	
work	2467 (82.71)
travel, medical or other	473 (17.29)
Result of urine drug tests in the past three months before transfer, *n* (%)	
positive	675 (22.96)
negative	1735 (59.01)
refused test	530 (18.03)
Duration of MMT before transfer (year), Mean ± SD	6.79 ± 2.03

**Table 3 ijerph-16-02023-t003:** The association between transfer service use and delayed return of transferred patients, obtained from the multilevel regression model.

Predictors	Model A: Return on Time or Delayed Return ^a^		Model B: Length of Delay ^b^	
	Univariable	Multivariable	Univariable	Multivariable
Transfer service use-level variables				
Days of each transfer to other MMT clinics	1.00 (0.99–1.00)		0.99 (0.99–0.99)	
Average daily methadone dosage during transfer period (mg/day)	1.00 (0.99–1.00)		0.99 (0.99−1.00)	
Reason for transfer				
travel, medical or other	Ref	Ref	Ref	Ref
work	1.47 (1.07–2.02) **	1.17 (0.89–1.52)	1.23 (1.13–1.35) ***	1.20 (1.07–1.35) **
Compliance rate during transfer period	0.17 (0.13–0.23) ***	0.16 (0.13–0.21) ***	0.26 (0.23–0.28) ***	0.26 (0.24–0.29) ***
Duration of MMT before transfer	0.92 (0.85–0.99) *	0.88 (0.81–0.95) ***	0.94 (0.89–0.99) *	0.89 (0.84–0.95) ***
Results of urine drug test in the past three months before transfer (%)				
negative	Ref		Ref	Ref
positive	1.11 (0.88–1.40)		1.15 (1.06–1.24) ***	1.10 (1.02–1.19) **
Individual-level variables				
Frequency of transfer service use (times/year)	1.05 (1.00–1.11)		1.06 (1.02–1.10) ***	1.06 (1.02–1.11) **
Dropout and then MMT re-enrollment				
no	Ref		Ref	
yes	1.35 (1.00–1.82) *	1.70 (1.21–2.41) ***	1.44 (1.14–1.83) **	1.78 (1.38–2.29) ***
Age of initiated drug use (years)	0.99 (0.97–1.02)		1.00 (0.98–1.03)	
Intravenous drug use, *n* (%)				
no	Ref		Ref	
yes	1.05 (0.64–1.72)		0.37 (0.93–2.01)	
Age	0.99 (0.96–1.01)		1.01 (1.00–1.03)	
Sex				
female	Ref		Ref	
male	0.69 (0.42–1.13)		0.73 (0.51–1.05)	
Employment status, *n* (%)				
full-time job	Ref		Ref	
unemployed or part-time job	1.30 (0.93–1.81)		0.87 (0.68–1.12)	
Education level, *n* (%)				
>high school	Ref		Ref	
≤high school	1.16 (0.78–1.72)		0.95 (0.71–1.28)	

^a^ Model A: Both univariable and multilevel logistic regression were conducted between transferred patients who returned on time and those who delayed return. Not significant (*p* > 0.10) variables in univariable analysis were not included in the multivariable models. Odds ratio and adjusted odds ratio with 95% confidence intervals are reported. Transferred patients with delayed return = 1, Transferred patients return on time = 0; ^b^ Model B: Both univariable and multilevel Poisson regression were conducted; the independent variable was the length of delay. Prevalence ratio and adjusted prevalence ratio with 95% confidence interval (CI) are reported; * *p* < 0.10; ** *p* < 0.05; *** *p* < 0.01.

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
