# Peer review of "Factors Associated with Compliance among Methadone Maintenance Treatment Transfers: Evidence from Audit Records at Clinics in Guangdong, China"

_ijerph, 2019, doi:10.3390/ijerph16112023_

Round 1

Reviewer 1 Report

Thank you for the opportunity to review ”Factors associated with compliance among methadone maintenance treatment transfers: Evidence from audit records at clinics in Guangdong, China”. This retrospective quantitative records-based study seeks to fill a gap in the literature that examines delays in re-engaging with the home clinic by users of MMT in Guangdong, PRC. The study is well considered and well implemented, and I commend the authors for identifying and exploring these issues. However, there are a number of presentation issue that must be addressed before the paper is accepted for publication. I address these below.

The paper clearly identifies its research question and variables, and the methodological analysis is straightforward and uncomplicated. The methods and analysis are transparent. However, perhaps because of the way English is used by the authors it is not clear how many MMT clinics there are in Guangdong (line 72); we know that six clinics participated in the present study, but are there others  which were invited that did not participate? This must be made clear; if there are others which did not participate, of course, the authors must consider the implications for the study in the limitations section.

English language in the paper is problematic, and must be brought up to a publishable standard. I have communicated with the editors, and they have assured me that the Journal editors will assist with that. This editing must include using standard terms (univariate instead of univariable, and the same for multivariable; line 125); “passion” must be, I think ‘Poisson’ (line 122); “Ethnical” must be ‘ethical’ (line 128); “administrate” must be ‘administration’ (line 203); there are others (for example, lines 202, 227, 238). The referencing style must be reviewed and made consistent with the Journal style. Apparently a referencing software was used (incorrectly) and revisions must be made at least to references 13,18, 20 and 28; there are others which need to be reviewed for stylistic conformity. Otherwise, the references are relevant and recent.

Another presentation issue is that tabular data are centred in columns; this is very difficult to read, and must be revised to conform to Journal style. (When the paper was uploaded Tables 2 and 3 break across pages—this is an editorial issue that should be addressed. I would also recommend other footnote symbols than $ and #, since these symbols have other meanings.)

The most important and useful aspect of this paper is the Discussion section, of course. In my view the first two paragraphs (lines 177-194) of the Discussion really belong in the Introduction section, since they don’t specifically address the data collected in the study, but rather form important background to the present study.  Lines 238-243 set out appropriate limitations to the study; it would be helpful if the authors identified environmental features of the MMT clinics in Guangdong and the users of these clinics that may differ from other MMT clinics throughout PRC (they identify privacy in order to find work, but surely there are others- if this is not available it could be a ‘further study’ recommendation). The authors have made creative use of existing data to establish that there is a problem, and the extent of the problem. The largest contributor to the model seems to be length of time in treatment prior to transferring. This suggests that not merely administrative issues, but the entire model of MMT delivery needs to be reconsidered. The existing model seems quite punitive, and the authors might advocate for a model of care that is oriented more to a public health approach that is responsive to the lived lives of the population in care. The authors, of course, will be sensitive to the political implications of such recommendations.

 The authors note that they may have missed factors. The only way to discover what is really going on is to ask the MMT users themselves. Given the scale and scope of the MMT population this would be a significant undertaking, but one of the recommendations of this quantitative study could—and in my view, should—be that a follow-up or companion study be done that interviews users who delay re-engagement in MMT after a leave. This would allow MMT clinics to refine their models of care, client management, administration and service delivery to improve adherence and management.

Author Response

May 28 ,2019

Dear Aurora Tao,

Thank you very much for your letter dated May 19,2019 regarding our manuscript entitled “Factors associated with compliance among methadone maintenance treatment transfers: Evidence from audit records at clinics in Guangdong, China” (Manuscript ID: ijerph-489375). We appreciate the opportunity to revise and resubmit. We have revised the manuscript based on your and the reviewers’ suggestions and comments (see our item-by-item responses blow).

Response to Reviewer 1 Comments

Point 1: However, perhaps because of the way English is used by the authors it is not clear how many MMT clinics there are in Guangdong (line 72); we know that six clinics participated in the present study, but are there others which were invited that did not participate? This must be made clear; if there are others which did not participate, of course, the authors must consider the implications for the study in the limitations section.

Response 1: We thank the reviewer for pointing out the need for clarification, which has been further clarified in the revised version of the manuscript (Line 76). By the end of 2016, there are 66 MMT clinics in the Guangdong Province[1].We invited all the 66 clinics to participants in our study. Six clinics with full audit records agreed to join our study. Among 60 clinics did not included in our study, some of clinics rejected our invitation and rest of them can not provide the full audit records from 2008 to 2016. All the clinics in Guangdong provided the same methadone maintenance treatment and transfer service for all the patients. Therefore, our result could be reference for other MMT in Guangdong Province. However, as the reviewer’s comments, not all the clinics participants in our research. We have clearly acknowledged this in the Limitation part (line240-242).

Reference:

[1] Chen, T.; Zhao, M. Meeting the challenges of opioid dependence in China: experience of opioid agonist treatment. Curr Opin Psychiatry 2019, 10.1097/yco.0000000000000509

Point 2: This editing must include using standard terms (univariate instead of univariable, and the same for multivariable; line 125); “passion” must be, I think ‘Poisson’ (line 122); “Ethnical” must be ‘ethical’ (line 128); “administrate” must be ‘administration’ (line 203); there are others (for example, lines 202, 227, 238).

Response 2: Sorry for the mistyping. We correct “Passion” into “Possion” (line 128); “Ethnical” into “Ethical” (line 135), “administrate” into “administration” (line 207); “exacted” into “extracted” (line 223). We further revised the whole paper to make it more informative and read more smoothly with the help of the MDPI editing company.

We disagree with the comments that we should change the univariable and multivariable into univariate and multivariate (line 132 and line 133). According to the Biostatistics: A Methodology for the Health Sciences (2004) and Katz MH (2003) [1-3], the difference between multivariate and multivariable regression can be summarized as follows:

Table 1 The differences between multivariate and multivariable regression

Multivariate regression

Multivariable regression

The number of independent variables

More than one

More than one

The number of dependent variables

More than one

one

In our study, we have two models with two dependent variables(delay return/ the length of delay),respectively.

1) First Model: dependent variable is the delay return. We conducted the logistic regression to explore the factors related to delayed return (Model A: dependent variable: delay return=1 vs, return on time=0).

2) Second model: dependent variable is the length of delay (days). The Possion regression was employed to explore the factors associated with the length of delay (Model B).

Reference:

[1] Driver, R. Biostatistics: a methodology for the health sciences, 2nd edition. Journal of the Royal Statistical Society Series a-Statistics in Society 2005, 168, 467-467, doi:10.1111/j.1467-985X.2004.02073_18.x.

[2]Katz, M.H. Multivariable analysis: A primer for readers of medical research. Annals of Internal Medicine 2003, 138, 644-650, doi:10.7326/0003-4819-138-8-200304150-00012.

[3]Hidalgo, B.; Goodman, M. Multivariate or Multivariable Regression? American Journal of Public Health 2013, 103, 39-40, doi:10.2105/ajph.2012.300897.

Point 3: The referencing style must be reviewed and made consistent with the Journal style. Apparently, a referencing software was used (incorrectly) and revisions must be made at least to references 13,18, 20 and 28; there are others which need to be reviewed for stylistic conformity. Otherwise, the references are relevant and recent.

Response 3: Thanks for your suggestion. We have changed the style of references according to the journal.

Point 4: Another presentation issue is that tabular data are centred in columns; this is very difficult to read, and must be revised to conform to Journal style. (When the paper was uploaded Tables 2 and 3 break across pages—this is an editorial issue that should be addressed. I would also recommend other footnote symbols than $ and #, since these symbols have other meanings.)

Response 4:We thank the reviewer for this good suggestion. We have followed the reviewer’s suggestion and re-arrange the tables accordingly. The footnote symbols were replaced by “a” and “b”.

Point 5: In my view the first two paragraphs (lines 177-194) of the Discussion really belong in the Introduction section, since they don’t specifically address the data collected in the study, but rather form important background to the present study.

Response 5: Thanks for your comments. We have rewritten the two paragraphs and added this important background information in the Introduction section, please see the Line 56 to Line 61.

Point 6: The largest contributor to the model seems to be length of time in treatment prior to transferring. This suggests that not merely administrative issues, but the entire model of MMT delivery needs to be reconsidered. The existing model seems quite punitive, and the authors might advocate for a model of care that is oriented more to a public health approach that is responsive to the lived lives of the population in care. The authors, of course, will be sensitive to the political implications of such recommendations.

Response 6: Thanks for your comments. We have written a new paragraph (line 247-264) to clearly express the policy suggestions from this research. Our findings of this research suggested to provide a more flexible and friendly transfer management system to patients in MMT. We suppose the authorities could provide convenient transfer service or take-home doses for patients to avoid missed doses and promote their adherence. Please see line 226 to line 243.

Point 7: Lines 238-243 set out appropriate limitations to the study; it would be helpful if the authors identified environmental features of the MMT clinics in Guangdong and the users of these clinics that may differ from other MMT clinics throughout PRC (they identify privacy in order to find work, but surely there are others- if this is not available it could be a ‘further study’ recommendation).

Response 7: Thank you for your suggestion, we totally agree. In our study, all the patients lived in the Guangdong province. However, we did not collect the data from the other provinces of China. For this reason, we are unable to point out the difference between Guangdong and other areas of China. Compared with other parts of China, Guangdong province has better economic and provides more jobs, which may affect the utilization of transfer service in the MMT clinics. In a further study, we will expand the subject of research to figure out how environmental features influence MMT service delivery (line 271-274). This part of the recommendation is also added to the Limitation.

Point 8: The authors note that they may have missed factors. The only way to discover what is really going on is to ask the MMT users themselves. Given the scale and scope of the MMT population this would be a significant undertaking, but one of the recommendations of this quantitative study could—and in my view, should—be that a follow-up or companion study be done that interviews users who delay re-engagement in MMT after a leave. This would allow MMT clinics to refine their models of care, client management, administration and service delivery to improve adherence and management.

Response 8: Thank you for your suggestion. We agree with the reviewer’s comments. In this research, we do not know what is really going on during their delay return period in this research. To fill this gap, we are planning to do a qualitative interview to figure out what is really going on during the delayed return period, what is the patients need when they transferred and how the administration can improve the MMT delivery model.

Reviewer 2 Report

This study of patients in methadone maintenance treatment (MMT) program examined factors associated with compliance surrounding treatment transfers. This is an important topic dealing with an important population and I think this could add value to the current literatures and implications for policy surrounding transfers. I had less concern with the study (method, results, and conclusions) but more concerns with the English language and style. I strongly suggest the use of an English language editing service as there were some sentences that needed minor adjustments and other that made little sense as they are currently worded. However, upon editing, this manuscript adds much and should be lauded.

Author Response

May 28 ,2019

Dear Aurora Tao,

Thank you very much for your letter dated May 19,2019 regarding our manuscript entitled “Factors associated with compliance among methadone maintenance treatment transfers: Evidence from audit records at clinics in Guangdong, China” (Manuscript ID: ijerph-489375). We appreciate the opportunity to revise and resubmit. We have revised the manuscript based on your and the reviewers’ suggestions and comments (see our item-by-item responses blow).

Response to Reviewer 2 Comments

Point 1: I had less concern with the study (method, results, and conclusions) but more concerns with the English language and style. I strongly suggest the use of an English language editing service as there were some sentences that needed minor adjustments and other that made little sense as they are currently worded.

Response1: On behalf of all the co-authors, I sincerely thank the editor and reviewer for all comments to help us to improve the quality and clarity of the manuscript. Thanks for your affirmation and encouragement. We have polished the manuscript with the aid of MDPI editorial service and have uploaded the editing certificate in the submission system.

Round 2

Reviewer 2 Report

This is much improved!